# Green Space Exposure and Obesity in the Mexican Adult Population

**DOI:** 10.3390/ijerph192215072

**Published:** 2022-11-16

**Authors:** Nabetse Baruc Blas-Miranda, Ana Lilia Lozada-Tequeanes, Juan Antonio Miranda-Zuñiga, Marcia P. Jimenez

**Affiliations:** 1Nutrition and Health Research Center (CINyS), National Institute of Public Health of Mexico (INSP), Cuernavaca 62100, Mexico; 2Research Center of Nutrition and Health, National Institute of Public Health, Av. Universidad 655, Col. Sta. Ma. Ahuacatitlán, Cuernavaca 62100, Mexico; 3Department of Actuarial Science, National Autonomous University of Mexico (UNAM), Mexico City 04510, Mexico; 4Department of Epidemiology, Boston University School of Public Health, Boston, MA 02118, USA

**Keywords:** green space, obesity, environmental health, Latino population, spatial epidemiology

## Abstract

Green space or natural vegetation may reduce obesity risk by increasing opportunities for physical activity or reducing stress and exposure to other pollutants. Obesity prevalence in Mexico is ranked among the highest in the world. However, research on the association between green space and obesity in Mexico is lacking. We used data from the National Nutrition Survey in Mexico (2018–2019), a nationally representative sample of Mexican adults. The analytical sample included participants between 20–59 years of age (*n* = 12,631). We assessed exposure to green space using a 30 m resolution Landsat satellite Normalized Difference Vegetation Index (NDVI) from 2018. Linear regression models examined associations between NDVI and body mass index (BMI), adjusting for confounders. The mean age of the study sample was 38 (SD 0.19) years. Participants living in areas with the highest green space exposure had the lowest education level (53.51%) and socioeconomic status (28.38%) and were located in central (33.01%), south (30.37%), and rural areas (21.05%). Higher residential exposure to green space was associated with a mean decrease in BMI of −1.1 kg/m^2^ (95% CI: −1.59, −0.68). This is one of the first studies in Latin America to suggest a protective association between green space and obesity among Mexican adults.

## 1. Introduction

Obesity is a serious public health problem and it is an important risk factor for non-communicable diseases with the greatest burden of morbidity and mortality in the world [1]. In Mexico, obesity prevalence is increasing. At a national level, between 2012 and 2018, the combined prevalence of overweight and obesity in the Mexican adult population increased by almost four percentage points (71.3% vs. 75.2%, respectively) [2]. If the prevalence continues, it is estimated that the cost of non-communicable diseases derived from obesity will reach 23 billion dollars in the next few years [3].

In recent years, there has been a growing interest in understanding the roles of obesogenic environments. Exposure to green spaces such as parks, forests, gardens, and public areas with natural vegetation provides an opportunity to reduce the prevalence of obesity through the practice of physical activity and stress reduction, among other mechanisms [4,5,6]. In addition, NDVI is the most commonly used way to objectively assess exposure to green space in epidemiologic studies [7,8,9]. It has been suggested that exposure to green space reduces the risk of several adverse effects on health [10]; since it promotes an active lifestyle [11]. Recent studies carried out with the population of the United States [12], England [13], Puerto Rico [14], Spain [15], and North of Mexico [16] have used geographic information systems (GIS) to investigate the relationship between green spaces through the Normalized Vegetation Difference Index (NDVI) and its effects on health [17,18,19,20].

Several studies have found that green space exposure provides health benefits by encouraging physical activity and thus reducing risks for obesity [9,21]. In an Australian study of children, spending more time in green spaces was associated with a reduced prevalence of obesity compared to spending less time in green spaces [22]. In a study in Lithuania, children who lived in neighborhoods with less access to green space had a higher risk of obesity compared to those who lived in neighborhoods with more access to green space [23]. However, many studies focused on the adult population are still lacking. One study in England found that adults living in neighborhoods with more access to green space were more likely to achieve the physical activity recommendation and less likely to be obese than those living in neighborhoods with less access to green space [24]. In a study from South Korea, the authors found that green space was associated with a lower prevalence of non-communicable diseases and lower rates of obesity [25]. Another study from Canada reported that higher greenness exposure was associated with decreased odds of obesity among women [26]. However, most evidence comes primarily from developed and Western countries [9].

No research has studied the association between green space and obesity risk in Latin America or developing countries. This is the first study in Mexico at the national level to assess the association between exposure to green space and its relationship with obesity among the adult population. We aim to study the impact of green space on an understudied population with a higher risk of obesity and cardiovascular disease. This study fills an important gap in the research literature, including focusing on a middle-income country and shedding light on potential public policies to increase exposure to green space and reduce obesity in the Mexican adult population.

## 2. Materials and Methods

### 2.1. Study Population

For this cross-sectional study, we used data from the National Health and Nutrition Survey (ENSANUT) 2018–2019. The analytical sample included participants between 20–59 years of age (*n* = 12,631), representing 59,710,238 Mexican adults in the expanded sample. We used this age range based on ENSANUT’s definition of the adult population as individuals 20–59 years of age [27]. ENSANUT is a national survey of the Mexican population. ENSANUT has become Mexico’s most important epidemiological tool for learning about the population’s current panorama of health and nutrition [2]. The target population of ENSANUT 2018–2019 was household inhabitants in Mexico.

Details of ENSANUT’s sampling and methodology have been previously published elsewhere [27,28,29]. Briefly, the sampling procedure included a randomized selection of households, stratified by clusters, from the National Household Sampling frame designed by the National Institute of Statistics, Geography, and Informatics (INEGI). A sample size was set for the urban and rural strata and the four regions (North, Center, South, and Mexico City) [27].

Data was obtained with prior informed consent and the protocol of ENSANUT 2018–2019 and was submitted and approved by the Research, Ethics in Research and Biosafety of the National Institute of Public Health of Mexico (INSP).

We excluded data for participants younger than 20 and 60 years old or older, who did not have a geocoded address in the INEGI microdata laboratory and with incomplete information from the ENSANUT, 2018–2019 sociodemographic or anthropometric questionnaire.

### 2.2. Exposure

We define green space as parks, forests, gardens, and public areas with natural green vegetation. NDVI values were used to estimate the green vegetation amount around the residential address of ENSANUT 2018–2019 participants. We overlaid the raster of NDVI on the participants’ geocoded addresses, and we assigned the value of NDVI to the participant based on the pixel their home is in. NDVI is the most widely used satellite-derived indicator of the quantification of green vegetation on the ground (in a range between −1 and 1) and was extracted through Google Earth Engine (GEE) [30]. Leaves absorb solar radiation in the visible photosynthetically active wavelength and scatter solar radiation in the near-infrared wavelength to avoid overheating. NDVI is based on the ratio of the difference between the near-infrared region and red reflectance to the sum of these two measures [30]. Negative values (approaching −1) correspond to water, values around zero correspond to barren areas of rock or sand, positive values represent grasslands, 1 indicates the highest vegetation index. NDVI has been used as a marker of exposure to green space in previous epidemiological studies [12,13,14,15,16]. For this study, we used Landsat 7 satellite for the summer (1 June to 1 September) from 2018 since NDVI reaches its maximum and highest level of geographic variation during the summer. We evaluated different buffer sizes (e.g., 270 m and 1230 m buffers). Our main results are based on the 270 m buffer, and we add sensitivity analysis for the 1230 m buffer. Further, we used the cover function in R with NDVI data from the summer of 2017 to impute missing values of NDVI in 2018 (1.014%), assuming that NDVI would not significantly differ in a year.

### 2.3. Outcome

Trained research assistants made home visits to measure body weight and standing height without shoes and in light clothing at each assessment three times using standard techniques through professional and duty-calibrated anthropometrical equipment [27,31]. An average of the three measures of weight and height was used for analysis. Our main outcome of interest was continuous BMI, defined by the World Health Organization as a simple general indicator of the relationship between weight and height [32].

### 2.4. Covariates

The covariates included at the individual level were age (years), sex (male or female), education (elementary and junior school or less, high school, bachelor’s degree or higher), and physical activity (not active vs. active). Physical activity was self-reported using the International Physical Activity Questionnaire (IPAQ) (being moderate-vigorous physically active at least 150 min/week or not) [33]. The socio-economic status classification was based on a principal component analysis of household characteristics and assets to create a tercile index, and we classified it as low, medium, or high. At the neighborhood level, we included the area (urban or rural) and region (North, Center, Mexico City, South) according to the place of residence of ENSANUT participants.

### 2.5. Statistical Analysis

Means and standard deviations (SD) were calculated for continuous variables, and proportions and 95% confidence intervals (95% CI) for categorical variables. Chi-square tests were performed between the NDVI and variables of interest. Linear regression models were used to examine the association between green space (NDVI) and obesity (BMI), adjusting for the previously described covariates. In addition, we conducted stratified analyses by urban and rural areas, socioeconomic status, and regions of the country. We considered statistical significance with a *p*-Value < 0.05. All analyses were performed considering expansion factors and adjusting for design effects with the *svy* module in Stata version 14.0 (Stata Tx Corporation, College Station, TX, USA).

## 3. Results

### 3.1. Sociodemographic and NDVI Characteristics

Our final analytical sample included 12,631 participating adults between 20 and 59 years old, representing 59,710,238 Mexican adults in the expanded sample. Most participants were women (57.8%); the mean age was 38 (SD 0.19) years. Of the total participants, 53.51% had a basic level of education, and 75.4% were obese. Most of the participants lived in urban areas (78.9%), distributed mainly in the central and southern regions of the country. Participants who lived in areas with the lowest exposure to green space (NDVI quartile 1) had a high prevalence of obesity (>70%), had higher socioeconomic status, higher education level, and were more likely to reside in urban areas compared to participants with higher exposure to green space. In contrast, participants who lived in areas with the highest green space exposure (NDVI quartile 4) had the lowest prevalence of obesity (72.14% vs. 76.95%), were more likely to reside in rural areas, and had lower socioeconomic status and education levels compared to participants with lower exposure to green space (Table 1).

Of the participants, 81.21% reported moderate and vigorous physical activity. Participants who lived in areas with higher green space exposure were more likely to be physically active than those with lower green space exposure (84.1% vs. 79.1%). (Table 1) Results were similar for the 1230 m buffer (see Appendix A).

The mean NDVI was 0.25 (SD 0.003) for Mexico in 2018, with greater values in the central and southern regions of the country, while the northern region presents the lowest values of NDVI (Figure 1). Similarly, rural areas had a higher mean NDVI than urban areas for all regions (0.43 (SD 0.006); 0.21 (SD 0.003), respectively) (see Figure 2).

### 3.2. Association between NDVI and Obesity (BMI)

Estimates from the unadjusted linear regression model (Model 1) showed that participants who lived in areas with the highest exposure to green space (quartile 4 of NDVI) had −1.1 kg/m^2^ lower BMI (95% CI: −1.59; −0.68) than participants who lived in areas with the lowest exposure to green space (see Figure 3) in a 270 m buffer. Results for the 1230 m buffer were similar (−0.96 kg/m^2^, 95% CI: −1.41; −0.52) (see Figure 3).

After adjusting for confounders (Model 2), results suggested that participants who lived in areas with the highest exposure to green space (quartile 4 of NDVI) had −0.84 kg/m^2^ (95% CI: −1.49; −0.20) and −0.54 kg/m^2^ (95% CI: −1.16; 0.08) lower BMI for both buffers (270 m and 1230 m respectively), compared to participants who lived in areas with the lowest exposure to green space (see Figure 3).

The results remained consistent when we included physical activity in the models (Model 3) the results remained consistent. Mainly, participants who lived in areas with the highest exposure to green space (quartile 4 of NDVI) had −0.74 kg/m^2^ (95% CI: −1.43; −0.05) and −0.34 kg/m^2^ (95% CI: −1.01; 0.33) lower BMI for both buffers (270 m and 1230 m respectively) (see Figure 3), compared to participants in the first quartile of NDVI.

Estimates from the unadjusted associations in the stratified analyses by urban and rural areas (Model 1) showed that participants who lived in rural areas with the highest exposure to green space (quartile 4 of NDVI) had −2.00 kg/m^2^ lower BMI (95% CI: −3.23; −0.78) than participants who lived in rural areas with the lowest exposure to green space (see Figure 4). Results for urban areas were attenuated (−0.70 kg/m^2^, 95% CI: −1.34; −0.06). After adjusting for confounders (Model 2), the highest exposure to green space (quartile 4 of NDVI) had −0.81 kg/m^2^ lower BMI (95% CI: −1.57; −0.05) compared to participants who lived in urban areas with the lowest exposure to green space (see Figure 4). The estimate for rural areas was attenuated, and the confidence interval included the null (−1.04, 95% CI: −2.30, 0.22). Lastly, results were further attenuated when we included physical activity in the models (Model 3) and the confidence intervals included the null.

Estimates from the unadjusted associations in the stratified analyses by socioeconomic status (see Figure 5) suggest that among participants in areas with low and medium socioeconomic status, those that lived in areas with the highest exposure to green space (quartile 4 of NDVI) had −1.39 kg/m^2^ lower BMI (95% CI: −2.26; −0.51) and −1.89 kg/m^2^ (95% CI: −2.70; −1.08) than participants who lived in areas with the lowest exposure to green space. Results in the fully adjusted models (Model 3) remained consistent, suggesting a strong negative association between green space and BMI (−1.91, 95% CI: −3.28, −0.53) and (−1.08, 95% CI: −2.13, −0.03). Lastly, results for participants in the highest socioeconomic status suggested no association since the confidence intervals included the null.

Finally, in models stratified by region (see Figure 6), we observe that in Mexico City, participants that lived in areas with the highest exposure to green space (quartile 4 of NDVI) had a stronger protective association between green pace and the risk of obesity, compared to participants in other regions of the country.

## 4. Discussion

This is the first study that provides evidence of the national association between green space and obesity in Mexico. Our findings suggest that higher exposure to green space was associated with lower BMI among Mexican adults aged 20 and 59 years after adjusting for age, sex, area, region, socioeconomic status, education level, and physical activity. The ENSANUT survey provides a generally representative sample of the Mexican adult population. Consequently, clinically relevant population estimates can be generated. The protective association between green space and the risk of obesity in the general Mexican adult population, as well as in stratified analyses, provides strong evidence for public health research, urban planners, and prevention strategies.

We found a protective association of green space exposure immediately around the household of participants (at a buffer of 270 m). Several studies have used different buffers to measure associations between green space and its effects on health [12,13,14,15,16]. Our results are in accordance with a study carried out in Spain in which the authors found a reduction in the risk of obesity among adults who lived in urban areas with the highest exposure to green space immediately around the household of participants (−0.18 kg/m^2^, 95% IC: −0.38; −0.01) [15]. The results for greenspace exposure immediately around the household may indicate that closer access to green space would yield higher opportunities for physical activity and/or recreational activities [12,13,14,15,16], which suggests a greater benefit for the health of the Mexican adult population from closer exposure to green space [32,33,34,35,36,37].

Previous studies have suggested protective associations between green space and the risk of obesity [38,39]. However, to our knowledge, no studies have focused on the Mexican adult population using participants´ residential addresses at a national level and satellite-based greenspace metrics. Results from this study are consistent with a cross-sectional study with adults ≥19 years conducted in the Netherlands [40]. The authors used the national health survey (Public Health Monitor 2012, PHM) to examine the association between NDVI and BMI and found similar results in terms of a reduction in obesity risk (OR = 0.88, 95% IC: 0.86; 0.91) for participants in the highest quintile of green space compared to participants in the lowest quintile (at a buffer of 300 m) [40].

As suggested by the ecological model and previous knowledge on the predictors of obesity risk [41], we adjusted for several potential confounders available at ENSANUT, such as socioeconomic status. Unlike other studies where higher socioeconomic status is associated with higher exposure to green space and lower BMI [12,13,14,42], findings in this study suggested that participants with low and medium household socioeconomic status may benefit more from green space. This could be due to differences in the population under study, where participants with low socioeconomic status may carry out tasks that imply more physical effort, such as construction, as opposed to sedentary jobs in the high end of the socioeconomic status range [43]. In addition, vulnerable rural areas in Mexico, such as the mountains or countryside/planting fields, are wild green spaces where the population might be subject to food insecurity and therefore have a lower BMI [44].

Results from this study contrast with others that have found positive associations between green space and BMI and others that found null associations. However, it is important to mention that the definition of “green space” used in each study varies according to the cultural context of each country, where access to green space may have different connotations depending on the perceived level of safety or quality of the green spaces [45,46]. This study shares the definition of “green space” with other studies that have found favorable effects on the population’s health [9,12,13,14,21,22,23,24,25,26].

Our study has several limitations. First, the study’s cross-sectional nature does not allow us to infer causality; however, due to the richness of ENSANUT data, we were able to adjust for important confounding variables. Second, NDVI does not capture types of vegetation or land use (such as land for agricultural purposes), and it is not possible to know whether participants have access to or use the green space. Third, BMI was considered a general indicator of nutritional status, but it is important to note that it does not differentiate the body composition. Fourth, due to the study’s cross-sectional nature, we were unable to examine the mechanisms through which green space could impact the risk of obesity.

Some strengths of this study are that, to our knowledge, this is the first research project to use the cutting-edge methodology to measure green space exposure around participants’ residential addresses at a national level in a middle-income country. In addition, the nature of the study population allows us to speak of representativeness at a national level. Similarly, the metrics used in this analysis have been widely validated in epidemiologic studies in Mexico and allow for comparison with other research projects.

## 5. Conclusions

In a nationally representative Mexican adult population, exposure to green space was associated with lower BMI, even after adjusting for important confounders. This project is among the first to show unprecedented associations between the environment and health in a developing country and creates new opportunities for environmental epidemiological research in the Mexican population. These results suggest the importance of generating and promoting green spaces throughout the national territory to promote health among the Mexican adult population. The findings of this study may shed light on the urban design of healthy and sustainable cities from a public health perspective.

## Figures and Tables

**Figure 1 ijerph-19-15072-f001:**
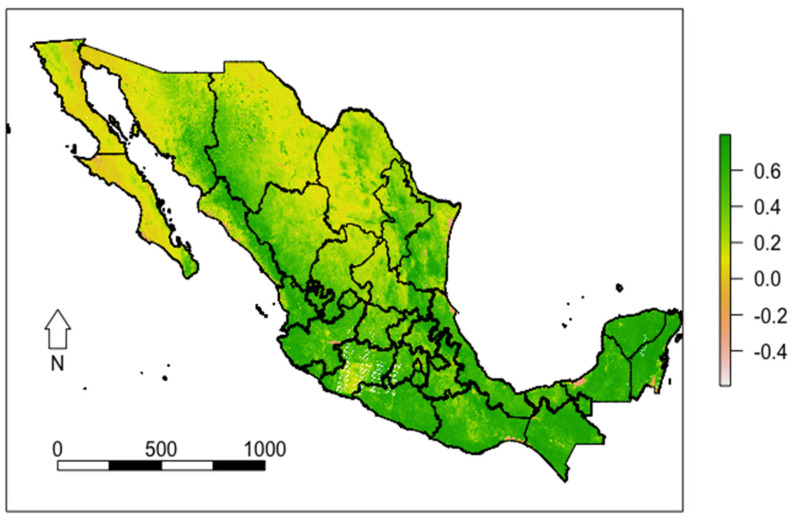
Normalized Difference Vegetation Index (NDVI) of Mexico 2018 values in summer (1 June to September).

**Figure 2 ijerph-19-15072-f002:**
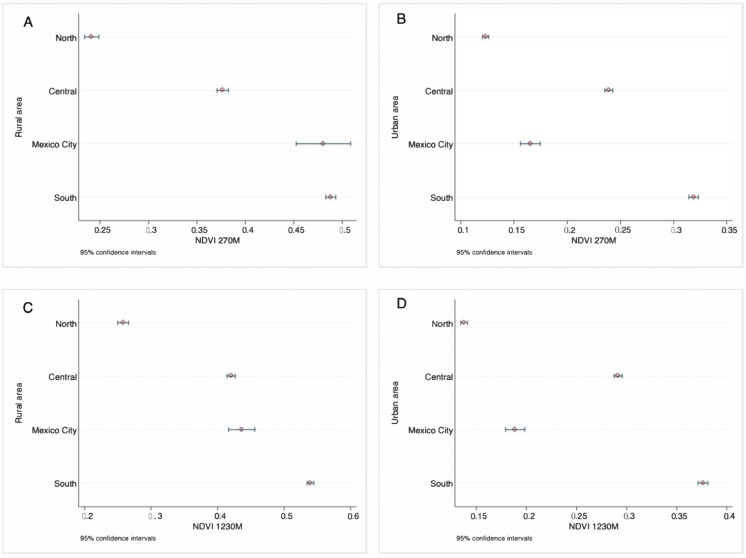
Distribution of NDVI by regions of Mexico during the summer of 2018, (**A**) buffer 270 m rural area, (**B**) buffer 270 m urban area, (**C**) buffer 1230 m rural area, (**D**) buffer 1230 m urban area.

**Figure 3 ijerph-19-15072-f003:**
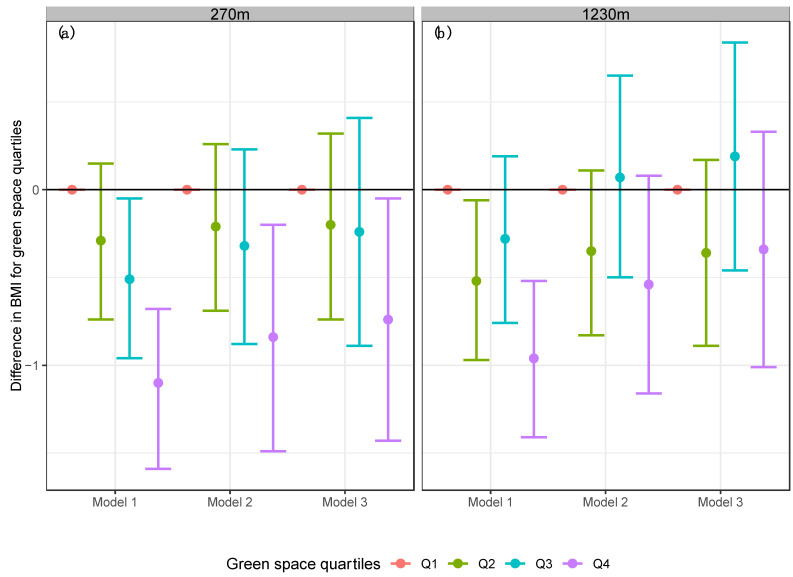
Linear models for BMI in the Mexican adult population by the difference quartiles of NDVI. (**a**) Buffer 270 m, (**b**) Buffer 1230 m. Model 1 = Linear regression model for BMI and NDVI. Model 2 = Adjusted linear regression model for BMI, NDVI, age, sex, area, region, socioeconomic status, and education level. Model 3 = Adjusted linear regression model for BMI, NDVI, physical activity, age, sex, area, region, socioeconomic status, and education level.

**Figure 4 ijerph-19-15072-f004:**
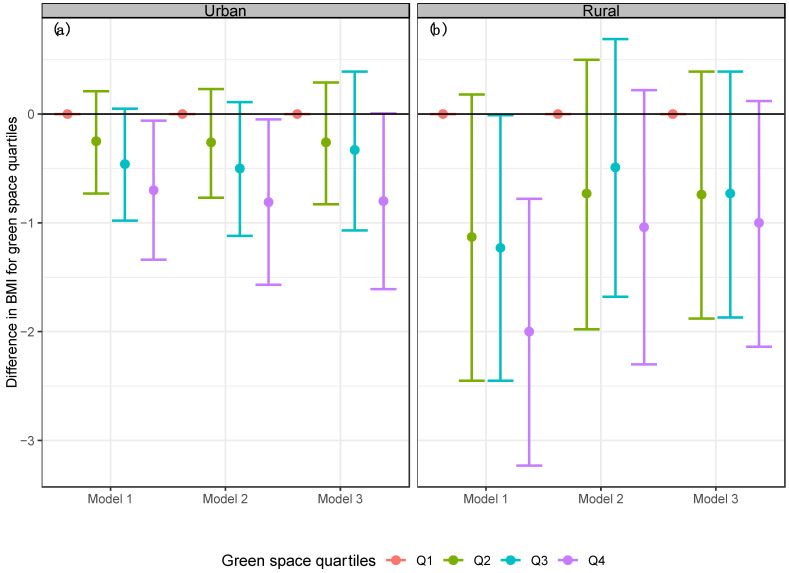
Linear models stratified by area for BMI in the Mexican adult population by the difference quartiles of NDVI. (**a**) Urban area (**b**) Rural area. Model 1 = Linear regression model for BMI and NDVI stratified by urban and rural areas. Model 2 = Adjusted linear regression model for BMI, NDVI, age, sex, region, socioeconomic status, and education level stratified by urban and rural areas. Model 3 = Adjusted linear regression model for BMI, NDVI, physical activity, age, sex, region, socioeconomic status, and education level stratified by urban and rural areas.

**Figure 5 ijerph-19-15072-f005:**
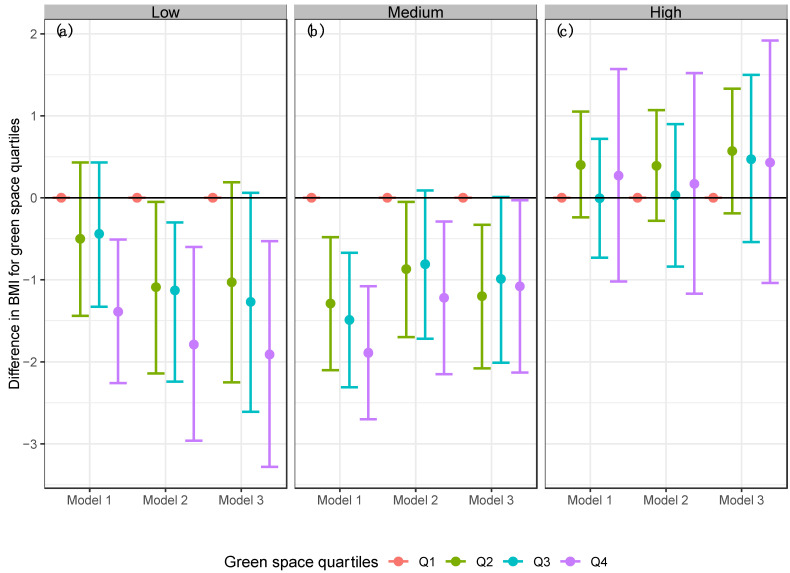
Linear models stratified by socioeconomic status for BMI in the Mexican adult population by the difference quartiles of NDVI. (**a**) Low, (**b**) Medium, (**c**) High. Model 1 = Linear regression model for BMI and NDVI stratified by socioeconomic status. Model 2 = Adjusted linear regression model for BMI, NDVI, age, sex, area, region, and education level stratified by socioeconomic status. Model 3 = Adjusted linear regression model for BMI, NDVI, physical activity, age, sex, area, region, and education level stratified by socioeconomic status.

**Figure 6 ijerph-19-15072-f006:**
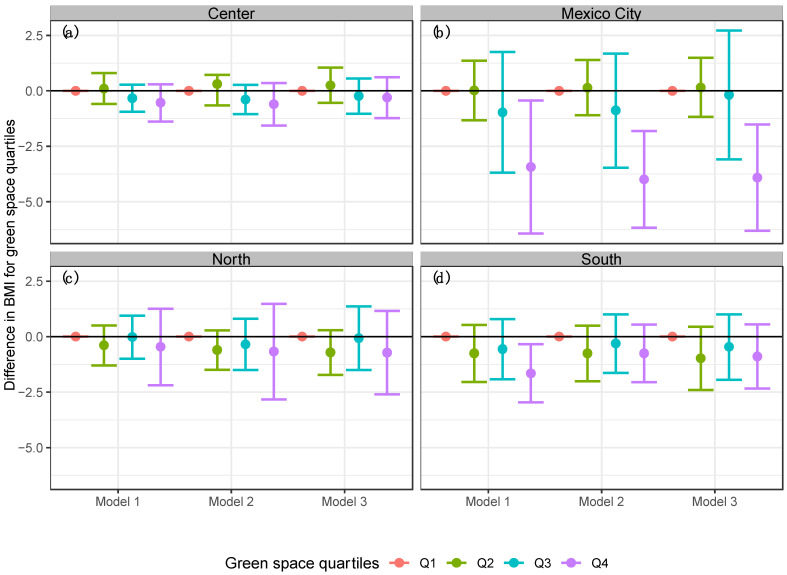
Linear models stratified by region for BMI in the Mexican adult population by the difference quartiles of NDVI. (**a**) Center, (**b**) Mexico City, (**c**) North, and (**d**) South. Model 1 = Linear regression model for BMI and NDVI stratified by region. Model 2 = Adjusted linear regression model for BMI, NDVI, age, sex, area, socioeconomic status and education level stratified by region. Model 3 = Adjusted linear regression model for BMI, NDVI, physical activity, age, sex, area, socioeconomic status and education level stratified by region.

**Table 1 ijerph-19-15072-t001:** Characteristic sociodemographic of the Mexican adults population aged 20 to 59 participants in ENSANUT 2018–2019, by quartiles of NDVI (buffer 270 m).

Characteristic	Total	NDVI (Buffer 270 m)			
*n* = 12,631 *** N = 59,710,238 ***		Quartile 1	Quartile 2	Quartile 3	Quartile 4
NDVI					
Buffer 270 m (mean ± SD)	0.25 ± 0.0027	0.10 ± 0.0011	0.21 ± 0.0012	0.33 ± 0.0018	0.51 ± 0.0033
Sex					
Male (%)	42.2	42.23	42.12	41.01	43.54
Female (%)	57.8	57.77	57.88	58.99	56.46
Age (years) (mean ± SD)	38.22 ± 0.19	39.14 ± 0.34	38.69 ± 0.38	37.49 ± 0.35	36.70 ± 0.37
BMI (mean ± SD) *	28.87 ± 0.08	29.27 ± 0.16	28.98 ± 0.16	28.76 ± 0.15	28.13 ± 0.15
Normal weight (BMI < 25 kg/m^2^) (%)	24.56	23.05	23.85	24.94	27.86
Obesity (BMI ≥ 25 kg/m^2^) (%)	75.44	76.95	76.15	75.06	72.14
Education level (%)					
<High School	53.51	44.83	48.48	57.01	72.25
High School Certificate	26.86	29.2	28.92	27.53	19.07
>High School, Bachelor’s degree or higher	19.63	25.97	22.61	15.46	8.67
Socioeconomic status (%)					
1st tertile (low)	28.38	12.07	18.83	35.62	62.96
2nd tertile (medium)	33.57	32.96	36.18	38.01	26.24
3rd tertile (high)	38.04	54.97	44.99	26.36	10.8
Region (%)					
Northeast	20.21	42.71	16.14	4.68	1.96
Central	33.01	21.79	38.89	46.54	30.48
Mexico City	16.41	29.61	16.83	6.54	2.75
South	30.37	5.89	28.14	42.23	64.8
Area (%)					
Urban	78.95	98.13	89.87	72.94	35.92
Rural	21.05	1.86	10.13	27.06	64.08
Physical activity moderate and vigorous, minutes/week ** (%) [*n* = 10,188 N = 47,451,492] ***					
Not Active < 150 min	18.79	20.85	19.9	16.52	15.89
Active > 150 min	81.21	79.15	80.1	83.48	84.11

Abbreviations: NDVI = Normalized Difference Vegetation Index; BMI = Body Mass Index; SD = Standard Deviation; * WHO classification; ** WHO guidelines on physical activity and sedentary behavior; *** *n =* sample size N *=* expanded sample; All *p*-value < 0.05.

## Data Availability

Not applicable.

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
