# Peer review of "Green Space Exposure and Obesity in the Mexican Adult Population"

_ijerph, 2022, doi:10.3390/ijerph192215072_

Round 1

Reviewer 1 Report

This is an interesting and important piece of work and builds upon a solid foundation of theory and similar research elsewhere. It is well written and there is ample additional data for the keen reader to delve deeper.

There is a clear outline provided and methodology appears sound.

I felt that the results might have been better presented than just as table – easier visually to absorb for readers with limited time.

Several other clarifications however need to be made before publishing:

-          Throughout the document, clarification of definition of green space/natural vegetation [13] [39] [126] is required. The definition of these is critical and becomes less clear again when utilising the NDVI derived data [75] – and critically explain a little about this data for those that might not understand – myself included. The issue I have here is of land-use and especially agriculture providing readings of ‘green’, as well as differentiations in vegetation cover between urban and rural areas and, critically, across different climatic regions. Or perhaps that is what the NDVI achieves?

-          This is the second point of clarification. How can the results from a randomly sampled population across the whole of a vast country (geographically) be comparable with such variation in ‘green’. i.e. There is so much variation is dependent upon aspect, latitude, altitude, land-use (again) etc.

-          What is the logic behind the buffer sizes [83] – unknown to me but an intriguing specific buffer size – imperial conversion perhaps?

-          [118] 12,631 participants is a staggering number -= could I please check that this is not a typo? Could I also confirm if this was just 20-59 yrs old, and critically, why this age group selected – literature to support why others excluded?

-          [121] I do not know if the sample set relates to existing averages across country – perhaps how this reflects broader demographic patterns and data (as confirmation)

-          [185 and others] – again definition of ‘green space’ becomes critical when reviewing the results and being able to support your findings.

-          [page 10] mention of USA and Europe – what might the NDVI recognitions be like in these areas however. Again, definition.

-          [last two paragraphs of Discussion] – these read as conclusion and might be better placed there.

Author Response

We thank the reviewer for their thoughtful feedback. We have followed your suggestions and think that our paper has improved based on your comments. Please see the attachment.

Reviewer 2 Report

Thank you very much for the opportunity to read and review the manuscript. The presented research on the relationship between the amount of natural greenery and obesity among the inhabitants of Mexico is very interesting and may have an international dimension.

However, there are a few inaccuracies in the content that would be worth explaining in order to improve the quality of the article.

1. The introduction section is too concise. In line 43 it says about research done so far in other countries. I think this research is worth describing. It will present what we know so far about the dependence of human health on the amount of greenery. In this section, it is also worth clearly presenting the purpose of the research. I think that "fills an important gap in the research literature" It seems to me that "fills an important gap in the research literature" is an insufficiently meaningful goal. With this assumption, the research would be conducted on the "art for art" principle, and it is not.

section 2.1. Study population

The number of people tested was not indicated (it is described below in the results but should be included in the methods section as well)

line 68. Why you exluded data for participants younger than 20 and over 60? Does it mean that all the forms carried out with participants in this age range were incorrect? Or maybe all those carried out in the age range between 20 and 60 were completed correctly? What did it result from?

line 77-79 It seems to me that the authors used mental shortcuts in these sentences. What was the rank of agricultural land and were they even taken into account? In the presented research, there is no categorization of green areas, and yet the NDVI index will be different for forests and different for arable lands. But can this affect obesity?

Section 2.3 Where did this data come from and how many people were tested?

I think that the article should present the obtained results using graphs and not only in tables.

Author Response

(The authors gave the same response as above.)
